# Effect of Pre-Surgical Orthopedic Treatment on Hard and Soft Tissue Morphology in Infants with Cleft Lip and Palate

**DOI:** 10.3390/diagnostics13081444

**Published:** 2023-04-17

**Authors:** Saki Ogino, Hitoshi Kawanabe, Kazunori Fukui, Ryoko Sone, Akihiko Oyama

**Affiliations:** 1Department of Dentofacial Orthopedics, Graduate School of Dentistry, Ohu University, 31-1, Misumido, Tomitamachi, Koriyama-City 963-8611, Fukushima, Japan; s-ogino@den.ohu-u.ac.jp; 2Division of Orthodontics and Dentofacial Orthopedics, Department of Oral Growth and Development, School of Dentistry, Ohu University, 31-1, Misumido, Tomitamachi, Koriyama-City 963-8611, Fukushima, Japan; k-fukui@den.ohu-u.ac.jp; 3Department of Plastic and Reconstructive Surgery, Fukushima Medical University, 1, Hikarigaoka, Fukushima-City 960-1295, Fukushima, Japan; okoyrenos@yahoo.co.jp (R.S.); akihiko.oyama@mac.com (A.O.)

**Keywords:** cleft lip and palate, pre-surgical orthodontic treatment, 3D imaging, nasoalveolar molding, one-stage surgery, 3D analyzer

## Abstract

The frequency of cleft lip and palate births in Japan is approximately 0.146%. The study aimed to compare the effects of NAM on restoring nasal morphology and improving extraoral nasal morphology in children with cleft lip and palate in the first stage of treatment using 3D imaging and oral model analysis. The subjects were five infants (37.6 ± 14.4 days old) with unilateral cleft lip and palate. The images taken with the 3D analyzer and oral model used for constructing the NAM at the first examination (baseline) and at the completion of the pre-surgical orthodontic treatment (157.8 ± 37.8 days old) were analyzed. The cleft distance was measured at the upper, middle, and lower points on the 3D images. On the model, the cleft jaw width at the maximum protrusion of the healthy and affected sides of the alveolar bone was measured. After the pre-surgical orthopedic treatment, the measured value on the model decreased significantly by a mean of 8.3 mm from baseline, and the cleft lip width narrowed by an average of 2.8 ± 2.2, 4.3 ± 2.3, and 3.0 ± 2.8 mm at the upper, middle, and lower points of the cleft, respectively. Pre-surgical orthopedic treatment using NAM can help narrow the width of the cleft jaw and lip. The sample size is stated at the study limit in the paper.

## 1. Introduction

Conventional treatment for cleft lip and palate consists of two-stage surgery with cheiloplasty and palatoplasty without pre-surgical orthopedic treatment. However, some medical facilities have used nasoalveolar molding (NAM) between cheiloplasty and palatoplasty as a pre-surgical orthopedic treatment [1,2]. Recently, some facilities have begun to perform cheiloplasty, palatoplasty, and gingivoperiosteoplasty (hereinafter, “one-stage surgery”) at around 5 months after completing pre-surgical orthopedic treatment to reduce burden and aim for early recovery, thereby helping the child’s development in suckling, language, and occlusion as well as optimizing aesthetics [3,4]. In primary treatment, pre-surgical orthopedic treatment using NAM is performed from the early postnatal period to around 5 months. Once the orthopedic treatment has reduced the cleft jaw width to around 2.0 mm after around 5 months, the one-stage surgery is performed. The one-stage surgery consists of cheiloplasty with the small triangular flap method, palatoplasty by Z-plasty and myoplasty, and alveoperiosteoplasty to promote development of the alveolar bone.

Conventional morphological assessment of soft tissues such as the lips and outer nose after cleft lip and palate treatment has generally used two-dimensional (2D) analysis, with some studies reporting three-dimensional (3D) analysis by using an impression model to observe morphological changes [5,6,7]. No studies to date have analyzed the facial morphology of children with cleft lip and palate using a non-contact 3D method. There are no reports of soft tissue changes in the cleft lip area.

Facial soft-tissue morphology assessments have used impressions taken with alginate materials; however, these methods have had problems, such as the inability to reproduce soft tissue morphology accurately due to pressure from the impression. Thus, we used the VECTRA_®_^H2^ 3D imaging system (3D analyzer; Integral Corporation, Tokyo, Japan) to take a contact-free equivalent of an impression to avoid both the negative effects of radiation exposure from X-ray imaging and the inaccurate measurements of models that represent displaced soft tissues caused by the pressure of impression-taking.

The aims of this study were to use a 3D analyzer and oral model analysis for infants with cleft lip and palate to investigate the effects of NAM in nasal morphological repair and the differences between images taken at the first examination (baseline) and after pre-surgical orthopedic treatment to validate the effects of one-stage surgery on infants with cleft lip and palate.

## 2. Materials and Methods

The subjects were five infants with unilateral cleft lip and palate (mean age, 37.6 ± 14.48 days) who received care at the Fukushima Medical University Department of Plastic Surgery.

Images taken with the 3D analyzer at baseline and after preoperative orthopedic treatment (157.8 ± 37.85 days old) and the oral model taken at the time of constructing the NAM were used.

Images were taken with the 3D analyzer (Figure 1a), and impressions were both taken by the same technician with the child in the supine position (Figure 1b).

3D analyzer images and impressions were taken after obtaining the informed consent of the parents, who were given explanations of the risks of impression-taking for constructing the NAM.

Cleft lip and jaw width measurements at baseline and after primary treatment were compared and analyzed.

As the subjects were infants, they were in the supine position and frontal and 45° images were taken from below. At the time of imaging, the light guide from the 3D analyzer was placed between the base of the nose and upper lip on the facial midline. The photographs were taken so that the guide focused from an approximate 30-cm distance (Figure 1b). To compensate for the bodily movements of the infant, 3D images taken in one direction were used for analysis (Figure 2) [8]. The cleft lip was divided into upper, middle, and lower points for measurement. The measurement taken at the upper point consists of the shortest distance between the end of the mucocutaneous junction on the medial side of the cleft and the end of the mucocutaneous junction on the lateral side of the cleft. The measurement taken at the middle point consists of the shortest distance between the point of maximum protrusion between the upper and lower points on the healthy and affected sides. The lower point measurement consists of the shortest distance between the lowest point of the lip on the healthy and affected sides. The affected side was defined as the point where the angle formed by the point on the mucocutaneous line on the oral commissure on the affected side, the peak of the Cupid’s bow on the affected side, and the mucocutaneous line on the affected side is equivalent to one-half of the angle formed by the oral commissure on the healthy side, the peak of the Cupid’s bow on the healthy side, and the midpoint of the Cupid’s bow mucocutaneous line on the affected side. The healthy side was defined as the point where the angle formed by the peak of the Cupid’s bow of the affected side that lies on the healthy side, the midpoint of the Cupid’s bow, the peak of the Cupid’s bow on the healthy side, and the midpoint of the Cupid’s bow is one-half of the angle formed by the oral commissure on the healthy side, the peak of the Cupid’s bow on the healthy side, and the midpoint of the Cupid’s bow (Figure 3) [9].

A digital dental caliper (Mitutoyo Corp., Kanagawa, Japan) was used to measure the maximum protrusion of the alveolar bone on the healthy and affected sides to measure cleft jaw width on the model (Figure 4).

Statistical analysis was conducted with the Mann–Whitney U test using SPSS 24.0 (IBM, Armonk, NY, USA). The significance level was *p* < 0.05.

The study protocol was reviewed and approved by the Ethics Committee of the Faculty of Dentistry, Ohu University (approval number: 338; approval date: 30 September 2021; facility number: 11000803).

## 3. Results

The cleft lip width measured at the upper point after pre-surgical orthopedic treatment using NAM decreased from baseline in all five cases (Table 1). Compared to the mean distance at baseline of 10.2 mm, the mean width after pre-surgical orthopedic treatment was 7.3 mm, demonstrating a significant reduction in cleft width (*p* = 0.016).

However, there was a variation in the amount of reduction, from 5.7 mm in one patient for whom NAM treatment was the most effective to a negligible 0.4 mm in the patient for whom it was the least effective. The mean reduction was 2.8 mm.

### 3.1. Changes in the Cleft Lip Width at the Middle Point after Pre-Surgical Orthopedic Treatment

The cleft lip width measured after pre-surgical orthopedic treatment using NAM decreased from baseline in all five cases (Table 2). Compared to the mean baseline cleft lip width measured at the middle point of 10.4 mm, the mean width after pre-surgical orthopedic treatment at the same point was 5.9 mm, showing a significant reduction in cleft lip width (*p* = 0.008). The largest and smallest reductions were 7.7 mm and 2.2 mm, respectively. The mean reduction in width measured at the middle point was 4.3 mm, which was larger than that measured at the upper and lower points.

### 3.2. Changes in the Cleft Lip Width at the Lower Point after Pre-Surgical Orthopedic Treatment

The cleft lip width measured at the lower point decreased in all 5 cases after undergoing pre-surgical orthopedic treatment with NAM (Table 3). However, the mean cleft lip width at the lower point after pre-surgical orthopedic treatment of 7.6 mm was not significantly reduced from the mean baseline width at the same point of 10.6 mm.In one patient, a large reduction of 7.7 mm was observed, whereas in another, the reduction was only 0.7 mm. The mean reduction was 3.0 mm.

### 3.3. Measurements of the Model

In comparison to the mean baseline cleft jaw width of 11.5 mm on the model, the mean cleft jaw width after pre-surgical orthopedic treatment significantly reduced to 3.1 mm (*p* = 0.008).

The mean reduction at the narrowest point of the cleft jaw was 8.3 mm. Since the NAM is constructed so that it narrows the cleft jaw width to approximately 2.0 mm, the width narrowed in all five cases and ranged between 9.9 mm in the patient with the largest reduction and 7.7 mm in the patient with the least change. The mean reduction was 8.3 mm (Table 4).

## 4. Discussion

### 4.1. Primary Treatment

The conventional three-stage surgery of cheiloplasty, palatoplasty, and gingivoperiosteoplasty requires exposing the child in infancy to general anesthesia three times. Previous research has warned of the toxicity to the central nervous system caused by general anesthesia drugs used in the surgeries that the infant is exposed to, as well as their effects on the developing infant [10]. Therefore, the one-stage treatment that we have adopted should reduce the physical burden on the child as well as save them from neurological harm (Figure 5).

Furthermore, unlike the conventional method, our one-stage treatment enables normalization of the oral cavity through surgery performed once at around 5 months. Moreover, the effects of NAM used in primary treatment allow the cleft jaw and lip to narrow, and performing the surgery at an early stage seems to prevent scar tissue and fistula formation after the one-stage surgery in all patients. Preventing cicatrization reduces the effects on the growth and development of the upper jaw, which should also minimize the effect on occlusion. Furthermore, preventing the formation of fistulas also helps reduce problems in articulation or the need to perform surgery to close the fistula.

### 4.2. Pre-Surgical Orthopedic Treatment

Grayson et al. [11,12] reported that narrowing of the cleft lip and jaw width can be accomplished by using NAM as pre-surgical orthopedic treatment, which is the method of pre-surgical orthopedic treatment that we use. Informed consent of the parents is obtained after explaining the risks of impression, after which a molding plate is created based on the impression taken. The NAM is used by placing tape with an orthodontic elastic onto the hook at the anterior palatal plate, and the tape is attached to the cheeks for fixation, which helps fit the NAM in the oral cavity such that a distal external force is exerted from the anterior of the palate (Figure 6 and Figure 7). Furthermore, soft resin is attached to the nasal stent to improve external nasal morphology [13].

In our study, both cleft lip and jaw width decreased significantly. The decrease in cleft lip width helps prevent invagination of the tongue during swallowing, which allows the NAM to narrow the cleft jaw. This is believed to help narrow the moveable mucosa attached to the alveolus and lips at the same time. Narrowing both the cleft jaw and lip width should thus effectively increase operability for the surgeon and prevent palatal fistula and cicatrization.

In the approximate 5-month period before the one-stage surgery, pre-surgical orthopedic treatment using NAM was provided. Using NAM for pre-surgical orthopedic treatment until the one-stage surgery prevents tongue invagination into the cleft jaw and lip. Intraoral normalization by NAM is expected, thereby improving suckling and promoting growth and development of the alveolar bone to closely resemble normality. The jaw cleft width after pre-surgical orthopedic treatment reaches approximately 2.0 mm, which should facilitate one-stage surgery and help prevent the formation of scar tissue and fistulation.

### 4.3. Method of Measurement with a 3D Analyzer and Differences with Conventional Measurement Methods

The 3D analyzer used in this study has high 2D resolution and enables taking high-definition magnified views of 2D images. Furthermore, the image data obtained from the imaging system is analyzed by the computer included in the system, which splits the camera lens for stereoscopic vision, where only one shot is necessary for a narrow field of view. Therefore, it is a suitable system for capturing a narrow field of view, such as when imaging the lips of an infant.

The principle of 3D image construction uses the dedicated Mirror 3D Analysis software (Integral Corporation) to approximate the values of the common parts of the image data by the method of least squares and creates a 3D model consisting of triangles (polygons) with sides of 1.2 mm in approximately 1 min. Color information is attached to the surface to complete a color image [14,15]. The digital data obtained can be applied to the measurement of the distance between two specified points, the volume, and simulations for various surgeries.

Very few methods of facial measurement to date have used 3D images but are rather based on facial impressions taken with impression materials after inducing general anesthesia. However, this involves the substantial effects of general anesthesia on young patients. The facial measurement method used in this study with the 3D analyzer has no risk of suffocation, unlike methods that involve taking impressions, and poses no physical burden of general anesthesia. As the 3D analyzer allows real-time observation of constructed 3D images from free angles, accurate measurements can be made from various angles. The only challenge is controlling the bodily movements and facial expressions of infants; however, this has been managed by taking multiple shots by one technician and aiming to minimize body movements and taking images with constant facial expressions as much as possible.

Computed tomography (CT) and magnetic resonance imaging (MRI) are also effective for observing internal craniofacial structures, including soft tissues, in a non-destructive manner. Recent studies report the use of CT to improve therapeutic outcomes in children with prenatally expected congenital abnormalities [16]. However, exposure to radiation is a health risk for both the mother and fetus. Strauss et al. note that although MRI is generally regarded as safe because there is no radiation exposure, the risks and benefits to the fetus have not yet been established [17]. Moreover, CT and MRI are time-consuming, making imaging infants difficult.

Therefore, the 3D analyzer is a more effective imaging modality for analyzing the orofacial morphology of infants with cleft lip and palate than CT or MRI, as imaging is possible in a short period of time and without radiation exposure.

### 4.4. Anatomical Relationship between Narrowing Cleft Jaw and Morphological Changes to the Lips

Fára et al. reported that in bilateral complete cleft palate, the orbicularis oris muscle, which runs horizontally from the oral commissure to the midline, cannot be crossed or connected at the midline due to the presence of the rupture. Rather, it reaches below the alar base on the lateral side of the rupture and below the base of the columella on the median side of the rupture, such that most of the muscle fibers stop at the periosteum of the maxilla, but some fibers end in the subcutaneous tissue [18].

In addition, in their study on the anatomy of the orbicularis oris muscle in cleft lip, Ross et al. reported that developing myogenic cells that encounter the margin of the rupture change directions and move upward to find a fixation site where they can stop and adhere to the bone and connective tissue. Should they not be able to find anchorage, the myofibrils would not differentiate into mature muscle cells. Ross et al. further noted the importance of surgeons attempting muscle reconstruction being aware of this [19].

Based on the above, pre-surgical orthopedic treatment to narrow the cleft jaw width causes perioral muscle fibers, such as the orbicularis oris muscle attached to the alveolar periosteum of the cleft jaw, to narrow the cleft lip width as it approaches the cleft jaw. As a result, narrowing both the cleft jaw and lip width is expected to facilitate first-stage surgery and improve postoperative stability.

Whereas an unaffected baby alternates between compressing and sucking in their nursing movement, babies with cleft lip and palate nurse with just compression without suckling pressure. Therefore, most infants with cleft lip, jaw, and palate do not generate negative oral pressure. In one study, the amounts of muscle activity on both the affected and healthy sides were significantly lower than those in healthy controls, and that of the affected side was significantly lower than the measured level, but it significantly increased postoperatively until the difference in maximum muscle activity was no longer significant [20]. This suggests that muscles that resemble normal anatomy and the closure of the cleft palate will improve suckling.

### 4.5. Limitations of the Study

Approximately 0.146% of Japanese babies are born with a cleft lip and palate [21]. The incidence is high worldwide, but the limited number of patients in this study and the limited number of patients eligible for one-stage surgery are limitations that prevent direct comparisons.

In the present study, the time to first-stage surgery was also long, at approximately 5 months, and the probability of the birth of a child with unilateral cleft lip and palate is low.

Furthermore, the importance of long-term follow-up for children who have completed one-stage surgery cannot be understated. Long-term follow-up is not always possible, especially when patients move or transfer to other hospitals or stop attending follow-up, preventing the study from continuing.

### 4.6. Future Implications

Narrowing of the cleft lip width at the upper, middle, and lower points and of the cleft jaw width as measured on the model after pre-surgical orthopedic treatment was observed. The greatest cleft width narrowing was observed in the order of the cleft jaw, followed by the middle, upper, and lower cleft lip, suggesting that the use of NAM and associated decreases in the cleft jaw and lip width enable easier one-stage surgery.

## 5. Conclusions

A successful treatment for narrowing cleft jaw and lip was suggested for infants with cleft lip and palate using pre-surgical orthopedic treatment with NAM.

The decrease in cleft lip and palate width suggested that natural lip and palate morphology could be achieved by implementing a pre-surgical orthopedic treatment.

As one-stage treatment may lead to simplicity and reduced physical burden for children with cleft lip and palate, the effectiveness of the treatment should be tested, despite the limited number of patients.

## Figures and Tables

**Figure 1 diagnostics-13-01444-f001:**
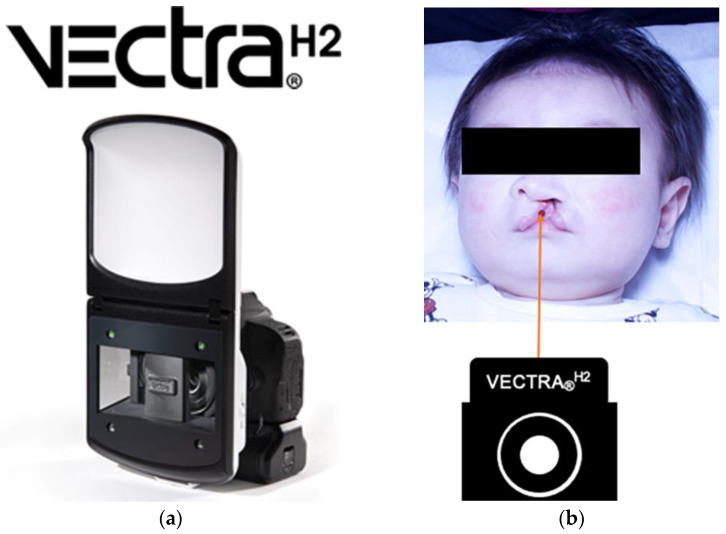
(**a**) Vectra^H2^ 3D analyzer (Integral Corporation, Tokyo, Japan). (**b**). Distance between subject and 3D analyzer.

**Figure 2 diagnostics-13-01444-f002:**
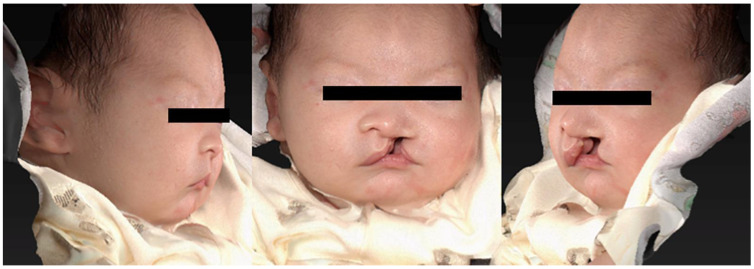
Three-dimensional imaging of one infant with the 3D analyzer.

**Figure 3 diagnostics-13-01444-f003:**
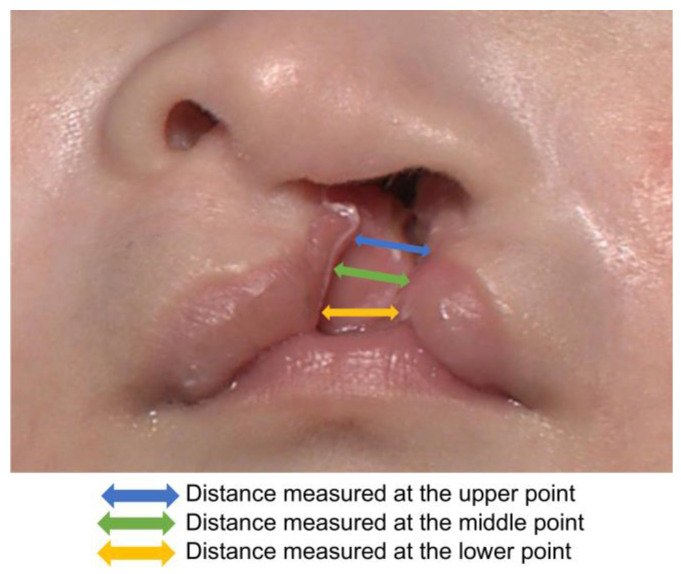
Width of the cleft lip measured on 3D images. Blue line, distance measured at the upper point; green line, distance measured at the middle point; yellow line, distance measured at the lower point.

**Figure 4 diagnostics-13-01444-f004:**
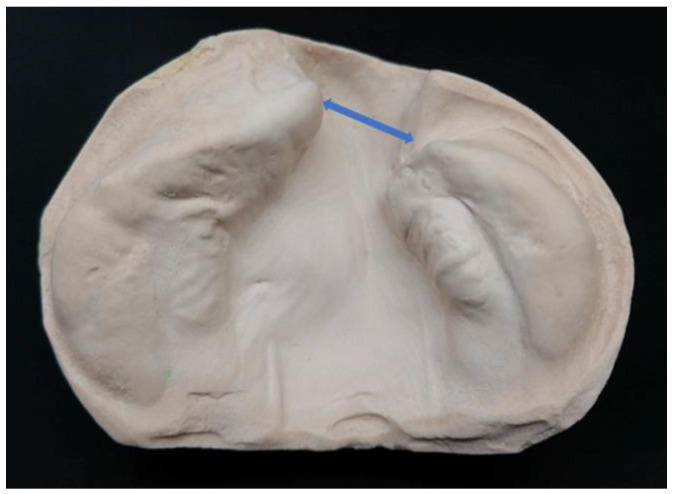
Points of the measurement of cleft jaw width. Blue line, cleft jaw width.

**Figure 5 diagnostics-13-01444-f005:**
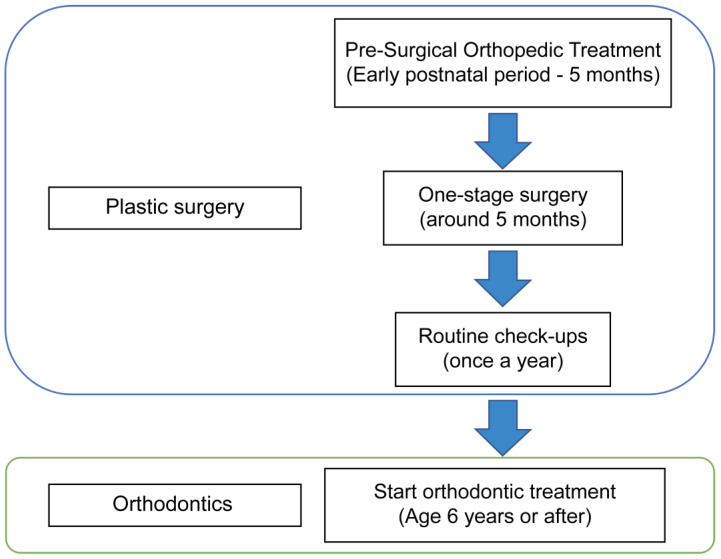
Flow of one-stage surgery.

**Figure 6 diagnostics-13-01444-f006:**
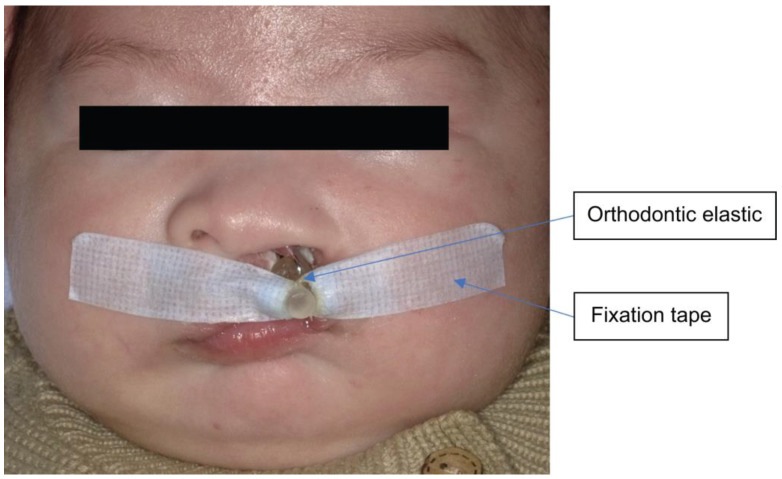
Facial photograph with nasoalveolar molding.

**Figure 7 diagnostics-13-01444-f007:**
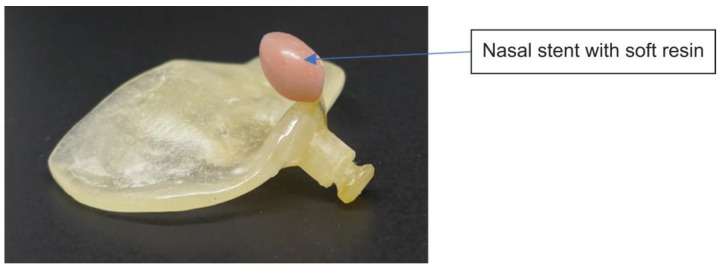
Nasal stent with molding plate used at our hospital.

**Table 1 diagnostics-13-01444-t001:** Comparison of cleft lip width between baseline and after pre-surgical orthopedic treatment (upper point).

Case Number	Baseline (mm)	After Pre-Surgical Orthopedic Treatment (mm)
1	13.1	7.4
2	7.7	7.3
3	10.7	6.3
4	9.2	8.1
5	10.4	7.3
Average	10.2	7.3
***p* = 0.016**		

**Table 2 diagnostics-13-01444-t002:** Comparison of cleft lip width between baseline and after pre-surgical orthopedic treatment (middle point).

Case	Baseline (mm)	After Pre-Surgical Orthopedic Treatment (mm)
1	13.3	5.6
2	9.9	7.5
3	8.8	3.0
4	8.4	6.2
5	9.8	6.3
Average	10.4	5.9
***p* = 0.008**		

**Table 3 diagnostics-13-01444-t003:** Comparison of cleft lip width measured at baseline and after pre-surgical orthopedic treatment (lower point).

Case	Baseline (mm)	After Pre-Surgical Orthopedic Treatment (mm)
1	14.6	6.9
2	11.0	9.7
3	8.6	4.8
4	9.3	7.8
5	9.5	8.8
Average	10.6	7.6
**Not significant**		

**Table 4 diagnostics-13-01444-t004:** Comparison of the baseline cleft jaw width on the model and after pre-surgical orthopedic treatment.

Case	Baseline (mm)	After Pre-Surgical Orthopedic Treatment (mm)
1	11.5	2.0
2	11.7	1.8
3	11.7	3.5
4	13.1	5.4
5	9.5	3.0
Average	11.5	3.1
***p* = 0.008**		

## Data Availability

Data are contained within the article. The data presented in this study are available in diagnostics.

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
