# Peer review of "Effect of Pre-Surgical Orthopedic Treatment on Hard and Soft Tissue Morphology in Infants with Cleft Lip and Palate"

_diagnostics, 2023, doi:10.3390/diagnostics13081444_

Round 1
Reviewer 1 Report
The paper "Effect of pre-surgical orthopedic treatment on hard and soft tissue morphology in infants with cleft lip and palate"describes the effects of NAM in nasal morphological repair. It presents a series of five cases treated in the Department.
In the Material and Method section, the institutional review number should be provided.
Figure 1 is unnecessary.
In tables 1,2,3 the difference between the measuring points and p values should be provided.
The conclusion should be more focused on the results.
MAny references are too old and should be revised.
Author Response
In the Material and Method section, the institutional review number should be provided.
Response :
Thank you for your valuable comment.
The institutional review number has been added in the Materials and Methods section of the paper (page 4, lines 118-120).
Figure 1 is unnecessary.
Response :
Thank you for your pertinent comment.
Figures 1 and 2 have been combined into Figure 1 as 1a and 1b (page 2, lines 71 and 74 ).
In tables 1,2,3 the difference between the measuring points and p values should be provided.
Response :
Thank you for your relevant comment.
These are described in the paper.
The conclusion should be more focused on the results.
Response :
Thank you for your insightful comment.
Additional information was added to the conclusion. “The decrease in the cleft lip and palate width suggests that the natural lip and palate morphology could be achieved by implementing a pre-surgical orthopedic treatment.” (page 10, lines 304-305)
Many references are too old and should be revised.
Response :
Thank you for your comment.
The references regarding this study are outdated because the 3D image measurement method is not available in previous literature.

Reviewer 2 Report
This paper evaluates the effects of pre-surgical orthopedic treatment with nasoalveolar molding (NAM) in infants with unilateral cleft lip and palate. The study based on five infants showed that the NAM treatment significantly reduced the width of the cleft jaw and decreased the width of the cleft lip.
Dear Authors,
you have correctly stated the limitations of this study. The conclusion that pre-surgical orthopedic treatment with NAM can significantly reduce the width of the cleft jaw and cleft lip is based on 5 cases. The small sample size, the long-term observation and assessment of functional outcomes, and the study conducted only in Japan have implications for the results, which may not be transferable to other populations with different genetic and environmental factors. This should also be mentioned in the abstract.
Figures 1 and 2 can be merged and positioned horizontally as Figure 1a and 1b (suggestion).
The work is interesting and suitable for publication.
Author Response
Response :
Thank you for your pertinent comment.
The small sample size has been stated as a study limitation in the revised paper.
Figures 1 and 2 can be merged and positioned horizontally as Figure 1a and 1b (suggestion).
Response :
Thank you for your comment.
Corrections were made to the placement of the figures.
The work is interesting and suitable for publication.
Response :
Thank you for your encouraging comment.

Reviewer 3 Report
Reasonable manuscript, though biased towards one method of presurgical infant orthopedics (NAM), without comparison of other methods with the exception of the comment that brief anesthesia is typically required for lip adhesion and latham placement which can be harmful at a young age. Also interesting to perform the repair at 5 months of age, later than many centers who opt for earlier repair. The limitations also leave out many limitations well-known with the use of NAM- frequent orthodontic visits for NAM adjustment are required, which can be prohibitive for low resource families or families who live far away. Gingivoperiosteoplasty also has shortcomings not discussed and there is no long-term follow-up for the patients in your study to see how the morphologic changes evolve over time. While I agree, this is a well-established appropriate and safe technique for cleft lip repair, there are pros and cons to all techniques and this was very biased to only one methodology. I would expand your limitation section and would also discuss the other potential treatment modalities often compared to NAM and perhaps discuss as a next step to compare NAM and the measurements taken in this study to other techniques using the same measurements to compare outcomes.
Author Response
Response :
Thank you for your valuable comment.
This will be an issue for future consideration.
